



# The Antarctic stratospheric Nitrogen Hole: Southern Hemisphere and Antarctic springtime total nitrogen dioxide and total ozone variability as observed in Sentinel-5p TROPOMI data

Adrianus de Laat, Jos van Geffen, Piet Stammes, Ronald van der A, Henk Eskes and J. Pepijn Veefkind

Royal Netherlands Meteorological Institute (KNMI), De Bilt, The Netherlands

Correspondence: Adrianus de Laat (laatdej@knmi.nl)

**Abstract.**

Denitrification of the stratospheric vortex is a crucial process for the Antarctic Ozone Hole formation resulting in  an analogous stratospheric "Nitrogen Hole". Here, 2018-2021 daily TROPOMI measurements are used for the first time for a

detailed characterization of this Nitrogen Hole. Nitrogen dioxide total columns exhibit strong spatiotemporal and seasonal variations associated with both photochemistry as well as transport and mixing processes. Combined with total ozone column data two main regimes are identified: inner-vortex ozone and nitrogen dioxide depleted air and outer-vortex air enhanced in ozone and nitrogen dioxide. Within the vortex total ozone and total stratospheric nitrogen dioxide are strongly correlated which is much less evident outside of the vortex. Connecting both main regimes are what is defined here as

"mixing lines", a third regime of coherent patterns in the total nitrogen dioxide column - total ozone column phase space. These mixing lines exist because of differences in three dimensional variations of nitrogen dioxide and ozone thereby providing information about vortex dynamics and cross-vortex edge mixing. On the other hand, interannual variability of nitrogen dioxide – total ozone characteristics are rather small except in 2019 when the vortex was unusually unstable. Overall, the results show that daily stratospheric nitrogen dioxide column satellite measurements provide an innovative

means for characterizing polar stratospheric denitrification processes, vortex dynamics and potentially long term monitoring if the total nitrogen column data record is extended with past satellite observations.

## 1. Introduction

Stratospheric nitrogen plays a crucial role in the formation of the Antarctic ozone hole. The hole forms during Antarctic springtime when halogens - mostly chlorine but also some bromine - are massively released from stable reservoir species

like $ClONO_2$, $HOCl$ and $HCl$ (Solomon, 1990; Solomon and Keys, 1992; Dessler, 2000; von Clarmann, 2013). Extremely low stratospheric temperatures during Antarctic winter after formation of the stratospheric polar vortex result in widespread formation of small particles containing nitrogen oxides – so-called Polar Stratospheric Clouds (PSC) – which slowly sediment. This process depletes the Antarctic stratospheric vortex from nitrogen oxides (denitrification/denoxification)



(Farman et al., 1995; Solomon and Garcia, 1983; Salawitch et al., 1989; Fahey et al., 1990; Tabazadeh et al., 2000; Weimer,
2022). The presence of the stratospheric polar vortex prevents resupply of nitrogen oxides from outside the vortex. The
return of sunlight to the Antarctic stratosphere during Antarctic spring to the then denitrified polar stratosphere leads to the
formation of halogen radicals (Solomon et al., 1999; Santee et al., 2008; Strahan et al., 2014). The lack of nitrogen oxides –
combined with the presence of PSCs – allows the halogen radicals to catalytically destroy ozone and thus cause the rapid
formation of the Antarctic Ozone Hole (Hurwitz et al., 2015). Stratospheric ozone depletion ceases either when all ozone is
destroyed or if the stratosphere becomes warm enough and unfavorable for PSCs allowing for the re-formation of halogen
reservoir species like HCl (Müller et al., 2008; Strahan et al., 2018; Stone et al., 2021). The warming in turn is caused by
increasing sunlight and absorption of that sunlight but sometimes also by stratospheric vortex instability.

The ozone depleted Antarctic stratospheric vortex is easily identified in for example satellite measurements of the total
ozone column (TCO3). It is characterized by a large gradient of ozone rich outer-vortex air and $O_3$ depleted inner vortex air.
This gradient is not only present in $O_3$ but also in other trace gases like nitrogen oxides, as outlined and pioneered by John F.
Noxon in the late 1970s (Noxon, 1978; 1979). This cliff-like large vortex edge trace gas gradient (Schoeberl et a., 1992;
Joseph and Lagras, 2002; Waugh and Polvani et al., 2010) is therefore referred to as "the Noxon cliff". The presence of this
cliff reflects air masses on either side of the cliff with very different chemical histories (Dirksen et al., 2011).

Studying Noxon cliff characteristics for a long time depended on numerical modelling (*e.g.* Solomon and Garcia, 1983;
Toon et al., 1987; Garcia and Solomon, 1994; Struthers et al 2004), ground based observations (*e.g.* Gil and Cacho, 1992;
Solomon et al., 1993; Kondo et al., 1994; Sanders et al., 1999; Struthers et al., 2004; Bortoli et al., 2005; Yela et al., 2005;
Cook and Roscoe, 2009) and aircraft or balloon measurement campaigns over Antarctica (*e.g.* Goldman et al., 1978;
Pommerau and Goutail, 1988; Fahey et al., 1989).

The advance of new innovative satellite instruments from the middle 1990s onwards but especially after 2000 enabled
exploration of new approaches for monitoring the Antarctic Noxon cliff (Bodeker et al., 2002; Ricaud et al., 2005; Manney
et al., 2006; Sato et al., 2009). The use of satellite observations for studying (mostly Antarctic) polar stratospheric nitrogen
compounds and the Noxon Cliff has been predominantly done with satellite limb observations (*e.g.* Callis et al., 1983; Mount
et al., 1994; Rinsland et al., 1996; Haley et al., 2004; Funke et al., 2005; von Savigny et al., 2005; Butz et al., 2006; Davies
et al., 2006; Kerzenmacher et al., 2008; Kühl et al., 2008; Kritten et al., 2010; Bourassa et al., 2011; Khosrawi et al., 2011;
Sofieva et al., 2012; Belmonte Rivas et al., 2014; Khosrawi et al., 2017; Dubé et al., 2020; Strode et al., 2022). Note that
many of these papers only touch upon the Noxon cliff, *i.e.* it is seen in the measurements and presented as an example of the
observational capacity of a certain satellite and/or data product. Some results have been reported on the use of nitrogen
dioxide ($NO_2$) total columns/stratospheric columns for nadir looking satellites but without a focus on polar regions
(Belmonte Rivas et al., 2014; Beirle et al., 2016). Note that a main interest in stratospheric or total $NO_2$ from nadir-viewing
satellites is because of the need to remove the stratospheric component from total column amounts to arrive at the
tropospheric $NO_2$ column (*e.g.* Hilboll et al., 2013).



There are a few research publications that touch upon satellite nadir total or stratospheric nitrogen dioxide ($NO_2$) observations over polar regions. Wenig et al. (2004) explore satellite nadir total stratospheric $NO_2$ (SNO2) column measurements from the GOME instrument. They identify the Noxon cliff in Arctic springtime observations in 1997 in 65 relation to the Arctic stratospheric vortex which persisted much longer than typical during that year. However, they do not explore the Antarctic region for similar purposes even though they mention multiple times that the Noxon cliff is present in both polar regions and that denitrification is larger over Antarctica relative to the Arctic. Richter et al. (2005) also explores GOME observations of total column $O_3$ (TCO3), total $NO_2$ (TNO2) as well as OClO over Antarctica during the early 2000s with a focus on the well-known September 2002 Antarctic vortex split (Ricaud et al., 2005; Richter et al., 2005; von Savigny 70 et al., 2005; Yela et al., 2005). They observe strongly reduced inner vortex SNO2 during early Antarctic spring that largely vanished after the vortex split. However, no effort is put into quantitively correlating TNO2/SNO2 with TCO3 and/or OClO. Adams et al. (2013) explore some OMI TNO2 data and TCO3 data in their study of ground-based observations at the Eureka station in northern Canada in relation to the anomalous longevity of the 2011 Arctic stratospheric vortex. They observe enhanced $NO_2$ and $O_3$ when outer-vortex air passes over Eureka associated with photochemical $NO_2$ production and the 75 stratospheric vortex preventing mixing of outer-vortex air with inner-vortex air, causing $NO_2$ and $O_3$ rich stratospheric air to accumulate in the region bordering the Antarctic stratospheric vortex. They also show the conjunction of $NO_2$ and $O_3$ depleted inner-vortex air in OMI data but do not analyze those observations in more detail. Gordon et al. (2020) explore OMI TNO2 and TCO3 in relation to (upper) stratospheric and mesospheric $NO_x$ formation due to energetic particle precipitation (EEP) but do not explore OMI TNO2 beyond that application. The Noxon cliff has also been identified satellite 80 nadir observations of nitrous acid ($HNO_3$) total columns of the European IASI satellite (Wespes et al., 2009, 2022; Ronsmans et al., 2016) as removal of $HNO_3$ from the Antarctic stratosphere is part of the denitrification process. Those studies showed that – not unexpectedly – Antarctic stratospheric vortex stability was important for inner-vortex $HNO_3$ and the strength of the Noxon cliff. However, in-depth analysis of the Noxon cliff in IASI $HNO_3$ observations is still also lacking. Note that the Noxon cliff has also been observed in nadir viewing satellite measurements of OClO (Kühl et al., 85 2006, 2008; Oetjen et al., 2011; Puķīte et al., 2021; Pinardi et al., 2022)

Satellite-observation-based exploration of the Noxon cliff and the denitrification process thus has almost exclusively been restricted to limb-sounding type satellite. Insofar as can be assessed the use of satellite nadir $NO_2$ measurements for in-depth studying the Antarctic stratosphere and denitrification has been absent. Even exploitation of the IASI $HNO_3$ data for this purpose has remained limited – in part also because of the need to average IASI $HNO_3$ data to reach sufficient data 90 accuracy.

Total $NO_2$ column measurements from a satellite instrument like TROPOMI nevertheless allows for studying stratospheric $NO_2$ especially since the entire Southern Hemisphere south of approximately 45° S – and thus Antarctica – is devoid of large $NO_2$ sources. Antarctica is effectively unpopulated and combined with a moratorium on industrial mining activities emissions associated with combustion are largely missing. Without much vegetated land, soil $NO_x$ emission are 95 small and although little is known about the occurrence of lightning near Antarctic the atmospheric conditions do not favor





widespread frequent occurrence of lightning. $NO_x$ production due to nitrate photolysis in the Antarctic snowpack is too small to yield tropospheric column amounts measurable by TROPOMI (France et al., 2011; Frey et al., 2013, 2015; Barbero et al., 2021). $NO_2$ emissions from the largest known single point source in Antarctic – the active volcano Mt. Erebus (Oppenheimer et al., 2005) – are likewise too small to affect $NO_x$ columns on a continental scale. Hence, TNO2 at high southern latitudes is

dominated by SNO2 columns. There thus is every reason to start exploring TROPOMI TNO2 or SNO2 over Antarctica – where they are nearly each other's equivalent - to characterize the Noxon Cliff for $NO_2$ as well as the denitrification/denoxification process.

This paper presents the first steps towards assessing high spatial resolution daily TROPOMI TNO2 and SNO2 Southern Hemisphere middle and high latitude measurements and in particular its relationship with TCO3. First, the TROPOMI SNO2

measurements are evaluated by comparison with ground based southern hemisphere and Antarctic SNO2 column observations. Daily and multi-day TROPOMI TNO2 measurements are then explored to characterize their spatiotemporal distribution and variability over and around Antarctica during local springtime. Subsequently daily TROPOMI SNO2 column measurements are collocated with daily TCO3 data. Similarities and differences in spatiotemporal distributions of both TROPOMI SNO2 and TCO3 are identified, analyzed and discussed. The origins of the complex relation between TCO3

and SNO2 in and around the Antarctic stratospheric vortex are briefly hypothesized and recommendations are provided about how satellite data of SNO2 columns could be further explored and used for studying stratospheric nitrogen.

## 2. Satellite data sources and data selection

### 2.1 TROPOMI stratospheric $NO_2$ data

The Sentinel-5 Precursor (S5P) satellite, launched on 13 Oct. 2017 in an ascending sun-synchronous polar orbit, with an equator crossing at about 13:30 local time, carries the Tropospheric Monitoring Instrument (TROPOMI; Veefkind et al., 2012). This instrument provides measurements in four channels (UV, visible, NIR and SWIR) of several atmospheric trace gases (such as $NO_2$, $O_3$, $SO_2$, HCHO, $CH_4$, CO) and of cloud and aerosol properties.

The TROPOMI $NO_2$ data retrieval is performed from the visible band (400−496nm), with a spectral resolution and

sampling of 0.54 nm and 0.20 nm, respectively, and a signal-to-noise ratio of around 1500. Individual ground pixels measure in the along-track direction 5.6 km (7.2 km prior to 6 Aug. 2019) and in the across-track direction 3.6 km at the middle of the swath, which increases to about 14 km near the edges of the swath. The full swath is about 2600 km wide, which means that TROPOMI achieves global coverage each day, except for narrow strips between orbits of about 0.5° wide at the equator.

The $NO_2$ retrieval process (van Geffen et al., 2022a, 2022b) uses the three-step approach introduced for OMI (Boersma

et al., 2007, 2011). First a Differential Optical Absorption Spectroscopy (DOAS) is applied to determine the slant column



density, the total amount of $NO_2$ along the effective light path from sun through atmosphere to satellite. A temperature correction is applied to correct for the temperature dependence of the $NO_2$ cross sections, based on collocated temperature profiles from ECMWF (re)analysis data. Then information on the $NO_2$ vertical profile shape taken from a chemistry transport model / data assimilation system (for TROPOMI: TM5-MP) that assimilates the slant columns is used to determine the

stratospheric vertical column density, symbolized hereafter by $N_v^{strat}$. The final step determines the tropospheric vertical column using appropriate air-mass factors (AMFs). The total vertical column density can be determined either from the sum of the two sub-columns or directly from the retrieved slant column – which of these total columns is the appropriate one depends on the application, as described in the Product User Manual (PUM; Eskes et al., 2022).

Since most of the $NO_2$ is located in the stratosphere, this study looks only at $N_v^{strat}$, the precision of which is estimated to

be approximately $2 \times 10^{14}$ molec.cm$^{-2}$ (3.3 µmol m$^{-2}$) in the data assimilation. The spatiotemporal variations in SNO2 are also seen in TNO2 and the geometric $NO_2$ column (*i.e.* the slant column divided by the geometric AMF, *i.e.* without any model information; *cf.* van Geffen et al. (2022a)), but not in the tropospheric column. TROPOMI $NO_2$ data is reported in SI units, *i.e.* in mol m$^{-2}$, where the conversion factor to the more commonly used unit molecules cm$^{-2}$ is $6.022 \times 10^{19}$ mol$^{-1}$.

The data used for this study comes from the version v2.3.1 intermediate S5P-PAL reprocessing (https://data-

portal.s5ppal.com/products/NO2.html; last access: 06 Dec. 2022) over the period 1 May 2018 up to 14 Nov. 2021, followed by the operational v2.3.1 and v2.4.0 processing. The latter version change has little to no impact on the stratospheric $NO_2$ column and can therefore be ignored in this study. For some info on the different versions, see van Geffen et al. (2022a), the Product ReadMe File (PRF; Eskes et al., 2021) and the latest PRF of the operational product (Eskes and Eichmann, 2022).

The stratospheric $NO_2$ column of all ground pixels with valid retrieval (qa_value > 0.50) of all 14 or 15 orbits of a given

day, *i.e.* orbit files with a start date & time in the file name for that day (irrespective of the actual sensing start and end), are arithmetically averaged on a $0.8° \times 0.4°$ grid (*i.e.* there are in total 450 by 450 grid cells globally). No weighting in space, time, or with measurement errors is applied. The daily gridded data is more convenient for various statistical analyses than using daily orbit data, for example for spatiotemporal averaging. We will return in the discussion section 4 to the question whether the gridding and averaging matters for the results presented here.

**2.2 TROPOMI stratospheric and/or total NO₂ column validation**

It is well established that nadir viewing satellite measurements of TNO2 are of good quality (Bortoli et al., 2013). An extensive first global validation of TROPOMI $NO_2$ can be found in Verhoelst et al. (2021). To highlight the quality of TROPOMI TNO2 data over and around Antarctica we explore TROPOMI data collocated with ground-based stations from the SAOZ network. The data is conveniently provided and visualized at the TROPOMI validation facility and the TROPOMI

validation server (https://mpc-vdaf.tropomi.eu/ & https://mpc-vdaf-server.tropomi.eu/). Extensive evaluation and reports are provided at the validation facility and server and in quarterly validation reports (Lambert et al., 2023) where also details about the SAOZ data can be found. We selected five Southern Hemisphere surface stations for comparing SAOZ sunrise data with TROPOMI SNO2 data from the TROPOMI offline data stream. These five stations are located inside and outside





of the vortex and also sample the vortex edge (Table 1). To account for the often large difference in solar local time between
the satellite (afternoon) and ground-based (twilight) observations, a diurnal cycle correction is applied based on model
calculations. The random error of SAOZ $NO_2$ total column measurements has been estimated at 4.7% with a total accuracy
of 5.9% (Hendrick et al., 2011). See Verhoelst et al. (2021) as well as the TROPOMI validation server for more details.

Figure 1A shows a time series of the comparison of TROPOMI stratospheric $NO_2$ data with the SAOZ observations at the
Antarctic site of Dumont d'Urville for the period 2018-2022. The Dumont d'Urville site is chosen as it is located
sufficiently far north to provide good sampling of the seasonal stratospheric $NO_2$ cycle while also sampling both inner and
outer Antarctic stratospheric vortex air during local spring. Note that observations are missing during the middle of winter at
Dumont d'Urville due to the polar night. Figure 1B shows the scatter plot of the same data.

Overall, the satellite measurements and ground-based measurements at Dumont d'Urville agree well (Table 1). The
correlation coefficient is 0.88 ($R^2$) with a bias of less than 2% and root means square differences of approximately 10%. The
regression coefficients equal 0.94 and almost 1.0 dependent on the regression method. The measurements at Dumont
d'Urville during Antarctic springtime sample both inner and outer vortex air as evidenced by the rapid changes between
large and small SNO2 values during springtime. The validation results for Dumont d'Urville thus cover a wide range of
atmospheric conditions. Results for the other four stations are rather similar (Table 1). Figures for the other four chosen
validation sites can be found in the appendix (Figures A2A and A2B) as well as on the TROPOMI validation server. The
validation results for all stations can be summarized as follows:

- correlations ($R^2$) are always better than 0.80 and up to 0.96
- biases are of the order of a few percent (11% for Rio Gallegos, Patagonia)
- root-mean-square differences vary between 10-20%
- standard errors are smaller than 1%
- regression values vary between 0.8 and 0.95
- results are fully consistent with Verhoelst et al. (2021)
- results are fully consistent with Lambert et al. (2023)

Given that the typical seasonal cycle and differences between inner and outer vortex air vary by a factor of two to five,
standard errors are a few percent or less combined with very high correlation coefficients and regression coefficients are
close to one, shows that single TROPOMI stratospheric $NO_2$ column measurements are of high quality and likely useful for
in-depth exploration of spatiotemporal Antarctic stratospheric $NO_2$ variability.



**2.3    Global ozone field data**

In this study assimilated TROPOMI NO$_2$ column pixel data is used and gridded at a spatial resolution much coarser than the original TROPOMI resolution while also averaging in time due to multiple polar overpasses per day as the main interest in this first exploratory study is at phenomena at continental scales. However, the TROPOMI NO$_2$ column pixel data itself are also already postprocessed level 2 observations, *i.e.* TROPOMI NO$_2$ data derived from a data assimilation system and in that sense not pure level 2 pixel data anymore. Hence, it was decided to compare the TROPOMI NO$_2$ data with gridded

assimilated TCO3 data rather than TCO3 data at orbit level. Furthermore, at the time of this analysis the MSR-2 TCO3 reanalysis dataset had not yet been extended in time to cover the entire period for which TROPOMI NO$_2$ data was available. Hence the TEMIS operational daily global assimilated TCO3 field is used here (Eskes et al., 2003; van der A et al., 2015; https://www.temis.nl/protocols/O3global.php, last access: 06 Dec. 2022) which is based on TCO3 level 2 data products of the GOME-2 instruments aboard the MetOp satellites (Munro et al., 2006, 2016). From this dataset the global total ozone

field at each longitude at local solar noon is used, which is close to the TROPOMI measurement time (section 2.1) and which is available for every day of the year for the full globe. The local solar noon ozone field is, for example, used for the operational TEMIS UV index and UV dose processing (van Geffen et al., 2017; Zempila et al., 2017). The local solar noon global TCO3 field is given at a longitude-latitude grid of 1.5°×1.0° and is re-gridded to 0.8°×0.4° to match the gridded NO$_2$ data. Note that differences between the TEMIS daily global assimilated TCO3 data and the MSR-2 TCO3 data are small.

GOME-2 has a 4 DU bias (MSR-2 has none) but otherwise GOME-2 and MSR-2 have similar root-mean square differences compared to ground observations (van der A et al., 2015). Hence, for the purpose of this study both datasets would be interchangeable. The question of whether using assimilated TCO3 data rather than collocated TROPOMI TCO3 orbit data will be discussed in section 4.

**3 Data analysis and results**

**3.1 spatiotemporal variability**

Figure 2 shows maps of the spatial distribution of SNO2 and TCO3 at local solar noon on 1 November 2018 as an example of daily data. In both panels a black line displays the TCO3 = 200 DU contour, a not uncommon reference value to mark the edge of the Antarctic ozone hole for the Southern Hemisphere polar vortex.

SNO2 depleted Antarctic inner vortex air and enhancement of SNO2 around the edge of the polar vortex are clearly

visible. Figure 3 shows the same data as in Figure 2 but for an Antarctic polar view and with a different color scale. There are clear similarities between the spatial patterns in SNO2 and TCO3. First of all, both show a significant reduction of values within the Antarctic stratospheric vortex. Secondly, values for both are strongly enhanced equatorward just outside of the vortex. Third, further equatorward of 45°S values of both start to decrease. And fourth: outside of the vortex values for both are reduced around 0° longitude and enhanced at the opposite side towards 180° longitude (wave-1 pattern).



However, there are also clear differences. The SNO2/TCO3 ratio for example does not show a clear vortex edge (Noxon cliff) like in both separate products. Furthermore, the gradient from the vortex edge towards the equator is smaller for TCO3 than it is for SNO2. These similarities and differences point to different processes governing their respective spatiotemporal variations: chemistry (sources & sinks) and stratospheric dynamics (source and sink regions and transport from sources to sinks).

Figure 4 shows the evolution of zonal averages of the SNO2 during the four Southern Hemisphere summers from 2018 to 2021, with the 200 DU ozone contour indicated by a black line. From these figures it is clear that the springtime SNO2 enhancement outside of the Antarctic vortex is kept out of the vortex during 2018, 2020 and 2021. The lack of such a well-defined SNO2 depleted area in 2019 is related to the weak Antarctic stratospheric vortex during spring 2019, which led to weak ozone depletion (Safiedinne et al., 2020; Wargan et al., 2020; Stone et al., 2021) and according to the TROPOMI NO$_2$
data thus also led to less denitrification. This is consistent with results found for IASI HNO$_3$ (Wespes et al., 2022).

**3.2 Correlating SNO2 and TCO3: 2D phase diagram**

Figure 5 displays TROPOMI TCO3 and SNO2 data for 1 November 2018 as a 2D histogram (phase diagram) revealing rather intricate patterns. For reasons explained below, the histogram was divided into three areas to be able to discriminate between the inner-vortex, outer-vortex and the vortex edge. For the area "MASK 1" SNO2 and TCO3 show a well-defined
linear relationship. The area is associated with the inner vortex and is characterized by small TCO3 values. The area "MASK 2" represents air outside of the vortex characterized by larger TCO3 values and somewhat larger SNO2 values than for the "MASK 1" area. Also, there is not such a well-defined linear relation between TCO3 and SNO2 for the "MASK 2" area as there is for the "MASK 1" area. The relation between TCO3 and SNO2 for "MASK 3" is much more intricate with what appear to be "coherent line structures" connecting the "MASK 1" and "MASK 2" areas. These "mixing lines" – by lack of
better expression - are found for both small and large TCO3 and SNO2 values. The largest SNO2 values are found in the "MASK 1/3" areas whereas the largest TCO3 values are found in the "MASK 2/3" areas. Note that the logarithmic color scale enhances the focus on parts of the distribution that are less frequent. There are thus essentially two populations: inside the vortex and outside the vortex. 16% of the histogram bins contain two thirds (~ 67%) of the data points and only approximately 10% of the data qualifies for MASK-3.

**3.3 multi-day periods and multi-annual data**

Two key follow-up questions are whether these results change significantly over time. Figure 6 shows phase diagrams similar to the one displayed in Figure 5 but for days combined during multiple day intervals (5-10-15-30 days) starting at 1 November 2018. Although this means that each panel covers a different time period, the results are nevertheless very consistent. The distinction of two clear concentrated populations and the "mixing lines" is present for each time period. The
results do also reveal a relation between TCO3 and SNO2 outside of the vortex albeit with a much larger spread. The high correlation between TCO3-SNO2 inside the vortex is also present during all periods. The distribution does shift towards



larger SNO2 values due to increasing SNO2 as part of the natural springtime SNO2 cycle. Similarly – albeit more difficult to distinguish in Figure 6, outer vortex TCO3 values become slightly smaller due to the natural seasonal springtime non-catalytic photochemical destruction of stratospheric $O_3$. However, for inner-vortex air the TCO3 distribution shifts towards

larger TCO3 values, reflecting the effects of dynamical mixing extra-vortex $O_3$-rich (upper) stratospheric air during late spring (de Laat and van Weele, 2011).

Figure 7 shows similar panels as in Figures 5 and 6 but for 5-day periods starting at the first day of each month from September to December 2018. The results are much more variable than for the multi-day differences highlighting the strong seasonality of especially Antarctic stratospheric inner-vortex TCO3 and SNO2. During early September $O_3$ depletion has yet

to commence. There are already two separate populations discernible but TCO3 values are still larger than 200 DU. SNO2 values are generally small, especially within the Antarctic vortex due to the denitrification process. Early October 2018 the catalytic $O_3$ destruction has strongly reduced stratospheric $O_3$. There is a group of datapoints with small TCO3 values and small SNO2 values. The air outside of the vortex is still characterized by large TCO3 values and still relatively small SNO2 values but larger values than during early September 2018. The "mixing lines" are also clearly discernible covering the

entire phase space between both main populations. The picture for early November 2018 is rather similar albeit that SNO2 values have further increased due to the natural seasonal cycle in SNO2. By early December 2018 the distribution is squeezed and values from both main populations are closer together. The vortex has largely disintegrated although remnants can still be discerned in TCO3 but interestingly enough not in SNO2 (see animation in the Supplementary Information). Remarkably the populations still cover the three previously defined areas "MASK1/2/3". This indicates that TCO3/SNO2

ratios are rather useful for characterizing the origins and locations of stratospheric air masses.

Figure 8 displays similar results as in Figure 7 for early October but for all years from 2018 to 2021. The results for 2018, 2020 and 2021 are very similar providing further support for the notion that the TCO3/SNO2 ratios can be used to characterize the origins and locations of stratospheric air masses. The results for 2019 are quite different, reflecting the unusually weak 2019 Antarctic stratospheric vortex. There are still two populations in 2019 albeit only weakly separated.

TCO3 and SNO2 values inside the vortex are larger compared to the other years. Overall, the anomalous 2019 vortex has a clear imprint on the TCO3 and SNO2 distributions. Note that during early September 2019 the amount of SNO2 depletion was still similar to those in 2018-2020-2021 (not shown). The normal vortex pre-conditioning during Austral winter 2019 thus was not unusual which is consistent with published analyses of the 2019 Antarctic springtime vortex (Wargan et al., 2020; Smale et al., 2021; WMO, 2022). The faster 2019 increase in SNO2 by early October compared to 2018-2020-2021

indicates that dynamics and mixing with - or influx of - $NO_2$-rich extra-vortex air is the main cause. Otherwise the SNO2 increase would have been slower and more in line with the other three years.

### 3.4 Qualitative explanation of phase diagram results

The consistency of patterns in the spatiotemporal variations in the TCO3-SNO2 distributions suggest some very basic underlying processes. For example, differences in the location of the Noxon cliff for TCO3 relative to the location of the





Noxon cliff for SNO2 should show up as patterns in the phase diagram. To provide a qualitative explanation of the observed patterns two simple series of longitudinal and latitudinal variations in TCO3 and SNO2 were created. For the first one, TCO3 and SNO2 vary as a sine wave along longitudes but with a different longitudinal phase (Figure 9). For the second one, TCO3 and SNO2 increased from pole to middle-latitudes and then decrease toward the equator to resemble the Noxon cliff but with a slightly different latitudinal change visually mimicking the observed TCO3 and SNO2 latitudinal gradients.

Figure 9 shows the results for the relation between both. For the phase-shifted sine wave functions, the results obviously show up as an oval. The latitudinal shifted results however follow a curve qualitatively not dissimilar from the observed "mixing lines". These results thus support the observation that the Noxon cliffs for TCO3 and SNO2 do not occur at the same locations which results in the emergence of "mixing lines" in the phase diagrams.

## 4 Discussion

The results presented here show that TROPOMI provides high quality daily SNO2 data for monitoring variations in SNO2 both inside and outside the Antarctic stratospheric vortex. It allows for studying the "nitrogen hole" - the denitrification process, as well as the "Noxon cliff" – the sharp gradient in trace gas amounts along the vortex edge, and associated seasonal changes during Antarctic springtime and interannual variability. Furthermore, combining the SNO2 data with high quality TCO3 data in phase diagrams reveals coherent patterns – "mixing lines" - linking the Antarctic

stratospheric air inside and outside the vortex.

A clear discrepancy was found between the location of the SNO2 Noxon Cliff and the TCO3 Noxon cliff. The few studies that provide information on the joint vertical distributions of $NO_2$ and $O_3$ suggest that the bulk of stratospheric $NO_2$ is found at higher altitudes than the bulk of stratospheric $O_3$ (Ridley et al., 1984; Lindenmaier et al., 2011). Differences in bulk heights which mostly determine total column variability link to differences in advection processes and might explain

differences in the location of the $NO_2$ and $O_3$ Noxon cliffs. Explorative studies using stratospheric chemistry models likely should help unraveling these issues. This in turn may contribute to developing applications and metrics for stratospheric $NO_2$-column-based Antarctic ozone hole monitoring. In addition, the notion of different bulk heights is consistent with the notion that the break-up dates or final warming of the Antarctic stratospheric vortex occurs later for lower stratospheric altitudes (higher pressure levels) (Butler et al., 2021; Lecouffe et al., 2022). For example, by late November 2018 there still

is well defined area with reduced TCO3 for which reduced SNO2 has already vanished (see animation in the Supplementary Information). A lower bulk height for TCO3 compared to SNO2 would mean that SNO2 anomalies would vanish earlier, as observed.

Furthermore, a strong inner-vortex correlation between SNO2 and TCO3 was found which was absent outside of the vortex. The SNO2 - TCO3 phase diagrams display a clear dynamical cycle reflecting springtime changes in chemistry and

dynamics. This cycle was consistently seen in multiple years (2018-2020-2021) but was significantly different in 2019, a year with a strongly perturbed Antarctic springtime vortex. Qualitatively the coherent patterns in the phase diagrams can be



explained by spatiotemporal differences in the phases of SNO2 and TCO3, *i.e.* where and when minima and maxima occur in SNO2 and TCO3. SNO2 and TCO3 are clearly not always and not everywhere *in sync*. This in part appears to be associated with the differences in Antarctic stratospheric denitrification and $O_3$ depletion. Denitrification is a wintertime

process starting already by early winter and causing the Antarctic stratosphere to be significantly depleted of nitrogen by the time sunlight returns (and thus TROPOMI starts to provide inner-vortex observations). The $O_3$ destruction cycle on the other hand critically depends on the presence of sunlight. At the start of springtime $O_3$ depletion has yet to speed up. During the month of September the amount of sunshine and duration of sunshine rapidly increase causing a rapid deepening of the Antarctic Ozone Hole. Hence, the denitrification and $O_3$ depletion cycles differ significantly in their timing. Similarly, the

results also revealed an earlier disappearance of the "$NO_2$ hole" relative the "Ozone hole", further supporting the notion that differences in chemistry and dynamics govern the differences in SNO2 and TCO3 behavior.

The observation of coherent spatial line structures ("mixing lines") in relation to stratospheric transport and mixing – including stratosphere-troposphere exchange - is not new. The presence of layered trace gas structures in the stratosphere (laminae, filamentation, contour advection (Waugh and Plumb, 1994; Newman et al., 1996; Appenzeller and Holton, 1997;

Orsolini and Grant, 2000; Who and Lagras, 2002)) is closely associated with the stability of the stratosphere, the conservation of potential vorticity and isentropic mixing (Waugh and Polvani, 2010). Stratospheric air masses often organize themselves in such long-lived laminae. For example, satellite observations of direct injection of volcanic material directly into the (lower) stratosphere have provided many examples of laminae development and filamentation because of the ability of satellites to observe sulfur dioxide and volcanic ash (Krotkov et al., 2021; de Leeuw et al., 2021; Kahykin et al., 2022).

Satellite observations of aerosols from wildfires have started to be used for similar purposes for stratospheric smoke (Khaykin et al., 2020; Magaritz-Ronen and Raveh-Rubin, 2021). And complex relationships between (long-lived) stratospheric trace gases have been used for understanding stratospheric dynamics (*e.g.* Hoor et al., 2002; Plumb, 2007; Barre et al., 2012; Hoffmann et al., 2017; Krasauskas et al., 2021). How exactly these processes and concepts relate to the observed "mixing lines" would make a relevant topic of future research.

Furthermore, model simulations could be used to assess (1) whether model simulations show similar phase diagrams and if so, (2) whether the model simulations contain clues for explaining the differences in spatiotemporal SNO2 and TCO3 behavior. The model simulations might also reveal caveats and missing processes in the model representation of stratospheric chemistry and Antarctic stratospheric vortex dynamics. In addition, the results can also be further explored towards a more thorough conceptual explanation of SNO2 variability. Comparison with IASI $HNO_3$ total columns might

help there as well, just like a comparison and evaluation of SNO2 with satellite $NO_2$ profile measurements from for example the OSIRIS, ACE-FTS or MAESTRO satellite instruments. A comparison with IASI $HNO_3$ could for example be used to explore whether both are more *in sync* than SNO2 is with TCO3. Comparison with limb satellite $NO_2$ profile measurements in conjunction with $O_3$ profile measurements should provide indications of which altitudes mostly determine column observations of $NO_2$ and $O_3$. In addition, evaluation of results from a different dynamical framework of the equivalent

latitude might help improve understanding of Antarctic stratospheric vortex edge dynamics. This links to the important





question of where and when vortex mixing takes place. It is well established that the Antarctic stratospheric vortex can be destabilized by increased extra-vortex wave activity. Direct observations of where and when that takes place is unclear but the combination of SNO2 and TCO3 (possibly extended with IASI HNO$_3$) might help identifying mixing regions.

An additional question is whether satellites other than TROPOMI that also measure SNO2 might help extend the SNO2 350 southern hemisphere record further back in time. A dataset going back to 2003 already exists via the QA4ACV NO2 data (Boersma et al., 2018). The combination of GOME (1995-2011), SCIAMACHY (2002-2012), OMI (2003-now), GOME-2 (2007-now), and OMPS (2012-now) potentially allows for reconstructing an almost 30-year record of Southern Hemisphere mid-latitude and Antarctic SNO2. Such a record could be probed for finding hints and clues of (Antarctic) stratospheric ozone recovery, as TCO3 is expected to change much faster due to decreasing O$_3$ depleting substances than SNO2 - mostly 360 due to emissions of N$_2$O and slowly increasing atmospheric N$_2$O concentrations (Struthers et al., 2004). Other processes relevant for Antarctic stratospheric NO$_2$ and O$_3$ are production of (upper) stratospheric NO$_x$ by energetic electron precipitation and by increased downwelling of upper stratospheric air by the expected speeding up of the Brewer-Dobson circulation (Gordon et al., 2020; Maliniemi et al., 2021; Müller, 2021).

Furthermore, there are some other aspects for further exploration. The validation could be extended to more ground-365 based comparison and more detailed evaluations. It could be assessed whether it matters if TNO2 is used (based on data assimilation) rather than SNO2. The assimilation SNO2 data is important for deriving tropospheric NO$_2$ but not necessarily the best estimate of TNO2. Note that there is no reason to assume that SNO2 and/or TCO3 data quality issues will change the findings of this paper, but only in-depth analyses will provide support for that assumption.

In this study gridded SNO2 was used to allow easy comparison with other data as Sentinel-5p data quality is still 370 improving and reprocessing of data is ongoing. A key question is whether results would differ for a Sentinel-5p pixel-level comparison of SNO2 and TCO3. A first brief assessment of using Sentinel-5p pixel-level comparison of SNO2 and TCO3 (see appendix Figure A3) yielded very similar results, indicating that results presented here are robust relative to using gridded data or pixel data or even data from different satellites.

Finally, limited use of nadir-viewing satellite measurements of NO$_2$ for studying the Noxon Cliff and the Antarctic 375 stratospheric denitrification process is somewhat surprising. The potential for their use to explore polar stratospheric chemistry and dynamics is evident from Wenig et al. (2004), Richter et al. (2005) and Adams et al. (2013). Satellite measurements of NO$_2$ - and tropospheric NO$_2$ column measurements - have been widely used for approximately two decades now. Total stratospheric NO$_2$ columns play an important role in deriving tropospheric NO$_2$ columns, as the stratospheric part needs to be removed from the total part (Boersma et al., 2003; Boersma et al., 2007; Boersma et al., 2018). This is typically 380 done by assimilating the satellite measurements of total NO$_2$ over clean regions into a numerical chemistry transport model to reconstruct the stratospheric column globally (Eskes et al., 2003; Boersma et al., 2004; Boersma et al., 2007). The assimilation therefore allows to determine the stratospheric column over polluted regions with sufficient accuracy and precision to subtract it from the total column to arrive at an accurate tropospheric column. This approach requires also sufficiently accurate measurements of stratospheric NO$_2$. Hence the quality of stratospheric NO$_2$ has for a long time been



assessed for various satellites (*e.g.* Boersma et al., 2004; Dirksen et al., 2011; Valks et al., 2011; Verhoelst et al., 2021; Lambert et al., 2023). Nevertheless, despite the fact that nadir stratospheric NO₂ column TROPOMI measurements turn out to be of very good quality their intrinsic value for stratospheric research has remained largely unrecognized.

## 5. Conclusions

This paper presents a first assessment of the use of Sentinel-5p SNO2 measurements for studying Southern Hemisphere
middle latitude and Antarctic stratospheric processes including the Antarctic Ozone Hole.

Comparison of gridded SNO2 and assimilated TCO3 via phase diagrams reveals intricate patterns. Three different regimes could be clearly identified: the inner vortex, the vortex edge and the extra-vortex region. Each regime is associated with its own SNO2/TCO3 characteristics. The vortex edge was characterized by so-called "mixing lines" in SNO2-TCO3 phase diagrams. A certain misalignment of SNO2 and TCO3 - different locations of the so-called Noxon cliff - was found
along the Antarctic stratospheric vortex pointing to vortex-edge dynamics as the root case. A possible explanation could be differences in bulk heights of SNO2 and TCO3 so that their respective total columns reflect processes occurring at different heights.

Springtime SNO2-TCO3 variations/changes are robust throughout single-day to multi-day statistics. Throughout spring the SNO2-TCO3 distributions change significantly as a result of chemistry and vortex dynamics including mixing of air
inside and outside the vortex. Regarding interannual variability the distributions are very similar for 2018-2020-2021 but significantly different from 2019 which was a year with an anomalously weak Antarctic stratospheric vortex and only weak O₃ depletion.

Seasonal changes in the phase diagrams indicate that both total column data products are sensitive to different heights and thus different processes. In general the vortex remains longer visible in TCO3 data than in SNO2 data. SNO2 is less
sensitive to the lower stratosphere– where the stratospheric vortex remains intact longer - than SNO2 so the nitrogen hole will disappear earlier than the ozone hole. Vertical tilting of the vortex edge combined with different vertical sensitivities likewise explains the presence of the third regime that in the phase diagrams linking the inner vortex regime with the outer vortex regime.

This study only presents a first glimpse of the great potential of high quality spatiotemporal satellite SNO2
measurements for studying stratospheric chemistry and stratospheric dynamics as well as long term changes in stratospheric composition extending the SNO2 record back in time in combination with for example the MSR-2 total ozone reanalysis (van der A et al., 2015). The ability to monitor stratospheric nitrogen is also more than welcome given that an important piece of stratospheric observational remote sensing capacity by way of the Microwave Limb Sounder (MLS) will end by 2025 or at the latest 2026 and no satellite missions are planned to fill the gap created by the end of the MLS mission.


*Author contributions.* J.d.L. wrote the paper and did the majority of data analysis and interpretation. J.v.G. did the data processing and participated also in the data analysis. P.S. is the instigator of this piece of research, P.V. is the PI of Sentinel-5p/TROPOMI, H.E. is responsible for the TNO2 data assimilation product and R.v.d.A. maintains the TCO3 data assimilation and data dissemination. All authors contributed to the discussion and interpretation of results.


*Competing interests.* The authors declare that they have no conflicts of interest.

*Data availability.* Data used in this paper is available via the ESA Sentinel-5p hub (SNO2), the TEMIS web portal (TCO3) and the TROPOMI validation data facility (SAOZ data and collocated TROPOMI TNO2 and TROPOMI assimilated SNO2)

https://s5phub.copernicus.eu/dhus/#/home

http://www.temis.nl

http://mpc-vdaf.tropomi.eu/index.php/nitrogen-dioxide/

*Acknowledgements.* Sentinel-5 Precursor is a European Space Agency (ESA) mission on behalf of the European Commission
(EC). The TROPOMI payload is a joint development by ESA and the Netherlands Space Office (NSO). The Sentinel-5 Precursor ground-segment development has been funded by ESA and with national contributions from The Netherlands, Germany, and Belgium. This work contains modified Copernicus Sentinel-5P TROPOMI data (2018-2022), processed in the operational framework or locally at KNMI.




**Figures**

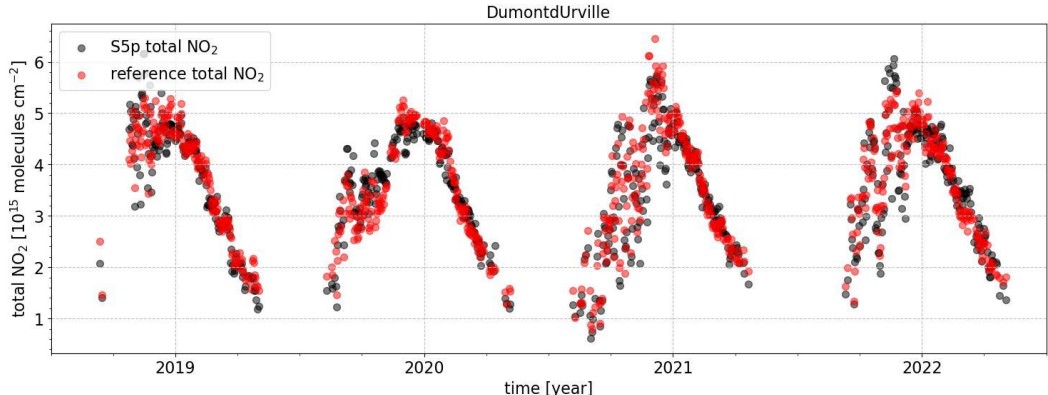

**Figure 1A.**

Comparison of S5p TNO2 and SAOZ sunrise TNO2 for the location of Dumont d'Urville. Data was directly obtained from the Sentinel-5p validation facility where also more details can be found about the SAOZ data as well as similar data visualizations (http://mpc-vdaf.tropomi.eu/index.php/nitrogen-dioxide/ accessed 21 November 2022).



**Figure 1B.**

Scatterplot of TROPOMI total NO₂ and SAOZ sunrise NO₂ as presented in Figure 1A. Regression coefficients are for an

ordinary linear regression and the orthogonal distance regression (ODR). Colors represent different times of the year (see

appendix Figure A1 for the corresponding colored version of Figure 1A)





**Figure 2.**

Maps of globally gridded TROPOMI-based stratospheric NO$_2$ (top; in μmol m$^{-2}$) and globally gridded local solar noon assimilated TCO3 (bottom; in Dobson Units or DU) on 1 November 2018. The location of the 200 DU ozone contour is indicated by a black line in both panels. Greys denote areas without TROPOMI data.




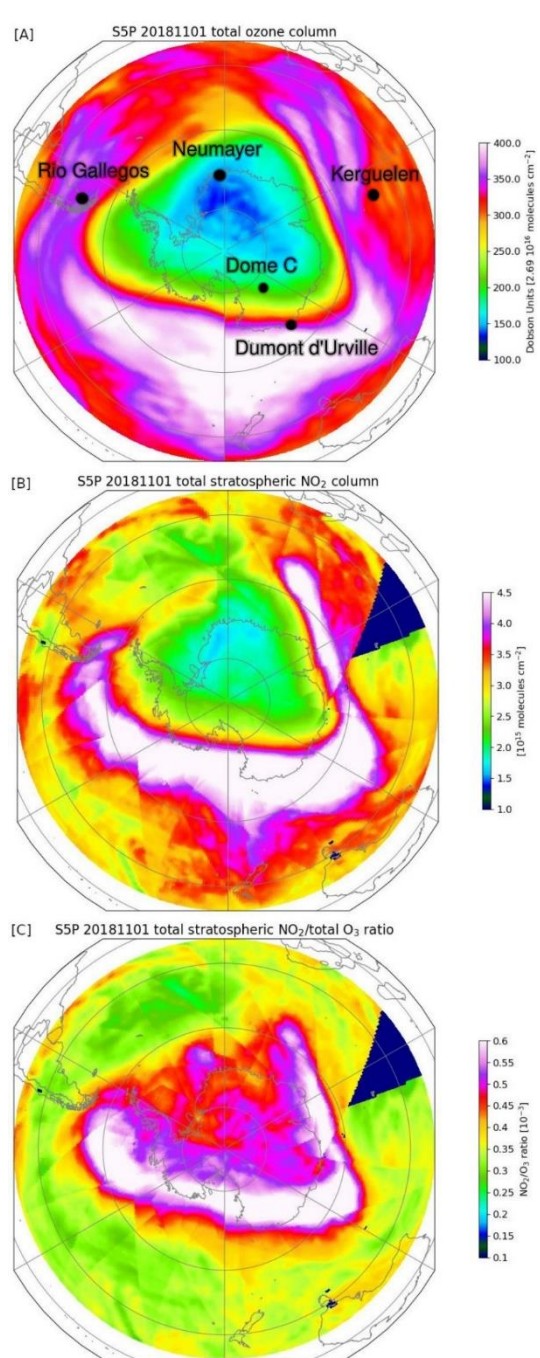

**Figure 3.**



As figure 2 (1 November 2018) but from an Antarctic polar view and with a different color scale. Panel C shows the

SNO2/TCO3 ratio of panels A+B.


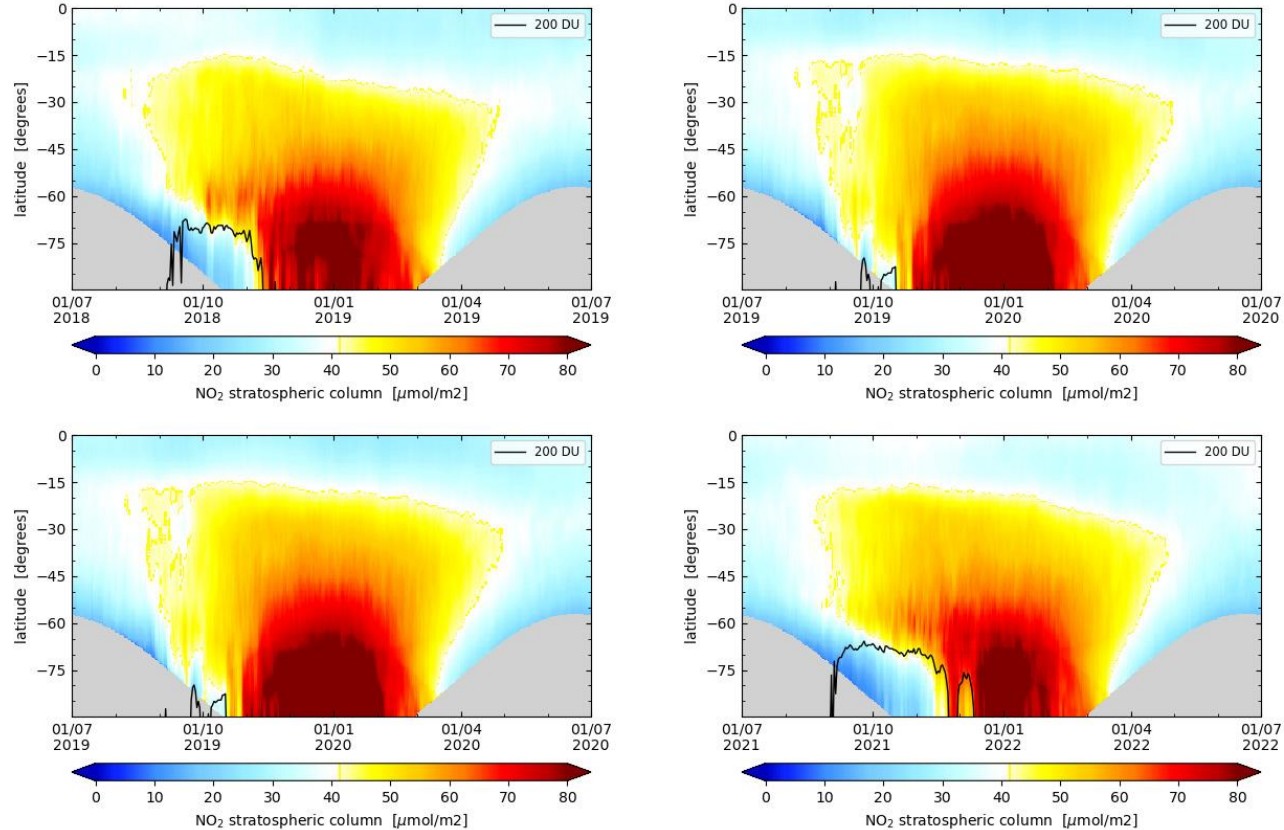

**Figure 4.**

Maps of the Southern Hemisphere daily zonal average SNO2 for four Southern Hemisphere summers (July-July) from 2018

to 2022. The location of the 200 DU ozone contour is indicated by a black line in all panels. Greys denote areas and times

within TROPOMI data.







**Figure 5.**

Left column: 2D histogram (phase diagram) of TROPOMI SNO2 vs assimilated TCO3 for 1 November 2018 and
corresponding spatial distribution of SNO2 and TCO3 as in Figure 3. The phase diagram is color coded according to the

logarithm of the number of counts. The phase diagram is a 100×100 pixel grid ranging between 0.0 - 6.0 $10^{15}$ molecules cm$^{-2}$

SNO2 and 0 - 500 DU TCO3. Right column: spatial distribution of TCO3 as in the lowest plot of the left column but filtered

on the masking in the phase diagram in the upper left plot.







**Figure 6.**

Phase diagrams of TROPOMI SNO2 and assimilated GOME-2 TCO3 Similar to the phase diagram in Figure 5 but for daily gridded data combining either 5, 10, 15 or 30 days starting at 31 October 2018.





**Figure 7.**

As Figure 6 but for 5-day periods starting at 1 September, 1 October, 31 October and 30 November 2018.





**Figure 8.**

As Figure 6 but for 1-30 October of each year between 2018-2021.



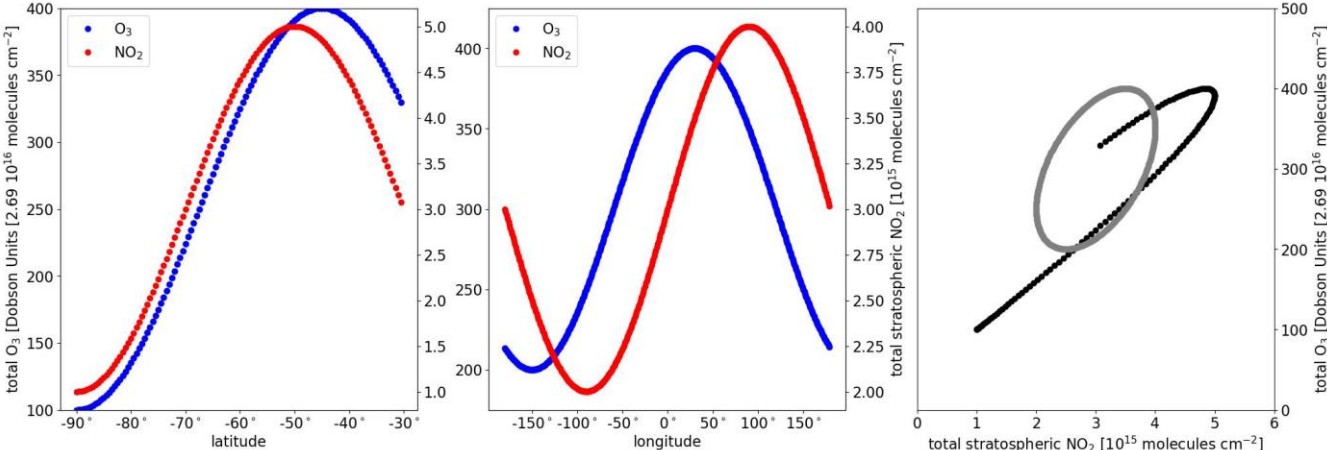


**Figure 9.**

Left panel: latitudinal sNO$_2$ and tO3 variations between 30S and 90S. The functions for sNO$_2$ and tO3 are slightly shifted in the latitudinal direction, with sNO$_2$ peaking earlier and decreasing faster towards the equator after the peak. The result of these two functions is indicated by the black line in the right panel. Right panel: longitudinal data and phase diagram of a

data series (sine wave) for sNO$_2$ (red) and tO3 (blue) with a longitudinal phase shift of 90 degrees. The amplitude of the sine wave is chosen to represent observed values but otherwise just a scaling factor. The result for these two functions is indicated by the grey line in the right panel.



**Tables**

| | $R^2$ [P] [S] | Bias $10^{15}$ [mean] [median] | Bias % [mean] [median] | RMS $10^{15}$ (err) | RMS % (err) | Fit [OLR] [ODR] |
|---|---|---|---|---|---|---|
| **Concorde Dome** (75.1°S, 123.35°E) | 0.834 0.821 | -0.035 -0.130 | -2.36 -3.64 | 0.466 (0.023) | 18.75 (0.93) | 0.808 0.875 |
| **Dumont d'Urville** (66.67°S / 140.02°E) | 0.884 0.882 | -0.039 -0.056 | -0.88 -1.67 | 0.371 (0.013) | 10.48 (0.36) | 0.938 0.999 |
| **Kerguellen** (49.35°S / 70.26°E) | 0.906 0.914 | -0.038 -0.065 | -2.51 -2.54 | 0.291 (0.009) | 10.23 (0.31) | 0.786 0.818 |
| **Neumayer** (70.65°S / 8.24°W) | 0.962 0.960 | 0.091 0.091 | 4.64 4.23 | 0.240 (0.012) | 14.01 (0.72) | 0.910 0.926 |
| **Rio Gallegos** (51.60°S / 69.32°W) | 0.925 0.925 | -0.282 -0.295 | -11.39 -11.62 | 0.244 (0.007) | 9.80 (0.27) | 0.899 0.944 |

**Table 1.**

Comparison of southern hemisphere and Antarctic SAOZ sunrise measurements of SNO2 (TNO2) with TROPOMI SNO2
observations. Correlations display the Pearson coefficient (P) and the Spearman coefficient (S). Fit coefficients are provided

for the ordinary linear regression (OLR) and the orthogonal distance regression (ODR).



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



**Appendix**

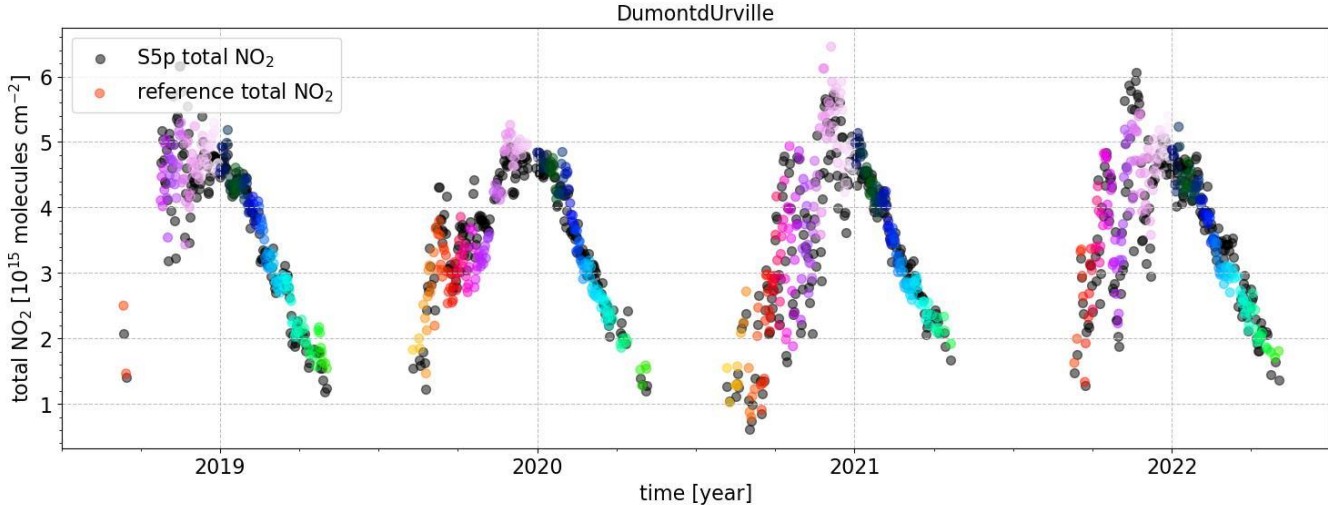

**Figure A1.**

As Figure 1A but color coded according to time of the year (color coding also used in Figure 1B).





**Figure A2A.**

As Figure 1A but for the other surface measurement stations in table 1.





**Figure A2B.**

As Figure 1B but for the other surface measurement stations in table 1.





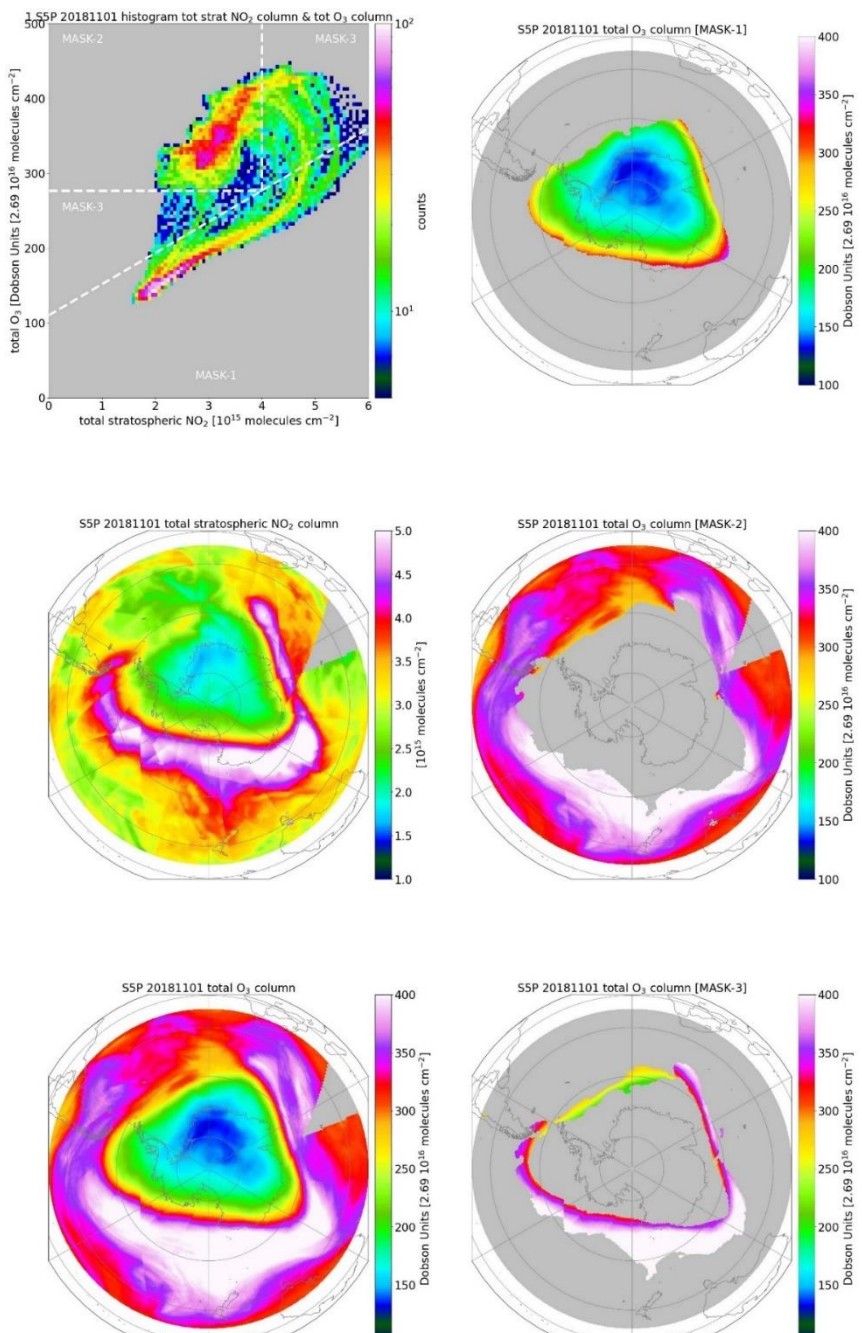

**Figure A3.**

As Figure 5 but for TROPOMI TCO3 pixel data collocated with TROPOMI SNO2 pixel data.