# Peer review of "The Antarctic stratospheric Nitrogen Hole: Southern Hemisphere and Antarctic springtime total nitrogen dioxide and total ozone variability as observed in Sentinel-5p TROPOMI data"

_EGUsphere, 2023_

## Referee Comment (RC2)

**General comments:**

"The Antarctic stratospheric Nitrogen Hole: Southern Hemisphere and Antarctic springtime total nitrogen dioxide and total ozone variability in Sentinel-5p TROPOMI data" provides a scientifically useful analysis of a TROPOMI measurements of nitrogen dioxide during the Antarctic ozone hole. The study demonstrates that co-located TROPOMI $NO_2$ and $O_3$ observations can be used to clearly identify the evolution of chemical differences between inner and outer polar vortex air masses throughout the springtime. While demonstrating the viability of a new dataset for the analysis of the Antarctic ozone hole is scientifically important, improvements to the presentation of the data and details about the design of the analysis are needed.

**Specific comments:**

1. The introduction could be improved by focusing on the advances offered by the TROPOMI dataset and the authors' analysis, specifically:
    a. What improvements or unique capabilities does this satellite dataset offer?
    b. What problem or scientific question does the dataset answer?
    c. The background on prior satellite studies could be condensed.
2. Additional details in the methods describing the range of latitudes used in each analysis are needed.
3. In the analysis of $NO_2/O_3$ correlations, how are the mask boundaries determined? What effects are there, if any, on the conclusions of the analysis if the boundaries are varied?
4. Can the "mixing lines" be geographically and temporally isolated into discrete eddies/filaments?
    a. Can the simple phase analysis of "mixing lines" recreate a similar structure on a 2-d plot? Figure 9 is not convincing on its own. Consider including actual seasonal trends.

**Technical comments:**

1. Subscripts for the common notation of $SNO_2$ and $TCO_3$
2. Improve the figure labels, especially dates
3. Figure 4 is redundant

---

## Author Comment (AC1)

Citation: https://doi.org/10.5194/egusphere-2023-2384-RC1

**RC1**

A Review of "The Antarctic stratospheric Nitrogen Hole: Southern Hemisphere and Antarctic springtime total nitrogen dioxide and total ozone variability as observed in Sentinel-5p TROPOMI data" by A. de Laat et al.

**General Comments**

This paper describes a new analysis of stratospheric "Nitrogen Hole" using TROPOMI nitrogen dioxide ($NO_2$) data and assimilated ozone ($O_3$) data. The analysis idea is somewhat new and found some new aspects on springtime cross-vortex chemistry/dynamics on $NO_2$ and $O_3$. However, since the stratospheric photochemical lifetime of $NO_2$ (10-100 s) is much shorter than that of $HNO_3$ (105-106 s), special care is needed to treat the stratospheric $NO_2$ data. The authors need to more carefully treat this point in the paper, as is pointed out in the following comments. When the modification of these points is completed in the paper, I think that the paper is worth published in Atmospheric Chemistry and Physics.

**Response:** we thank the referee for the review efforts and the comments that have helped improve the paper. Below follows a detailed response to the comments including for each comment a description of the modifications that have been made.

**Major Comments**

1) P.6, L.160: The authors first tried to validate the TROPOMI $SNO_2$ data with ground-based SAOZ data. However, the TROPOMI $SNO_2$ data are acquired at 13:30 local time, while SAOZ data are acquired at local sunrise. The authors claim that "a diurnal cycle correction is applied based on model calculations". Since this is a critical point for comparison, more detailed description is needed for this "diurnal cycle correction".

**Action:** we added this literal quote from Compernolle et al. [2021] that is also in Lambert et al. [2023]:

"the SAOZ measurements are adjusted to the TROPOMI overpass time using a model-based factor. This is calculated with the PSCBOX 1D stacked-box photochemical model (Errera and Fonteyn, 2001; Hendrick et al., 2004), initiated with daily fields from the SLIMCAT chemistry transport model (CTM). The amplitude of the adjustment depends strongly on the effective SZA assigned to the ZSL-DOAS measurements; it is taken here to be 89.5°. The uncertainty related to this adjustment is of the order of 10 %. To reduce mismatch errors due to the significant horizontal smoothing differences between TROPOMI and SAOZ measurements, TROPOMI $SNO_2$ values (from ground pixels at high resolution) are averaged over the air mass footprint where ground-based zenith-sky measurements are sensitive."

2) If ascending node crossing local time of Sentinel-5P is 13:30, the descending node crossing local time is 01:30. However, there is no description on whether the authors are using only ascending part of the orbit, or using both ascending and descending parts (full parts) of the orbit. Since the measurement local time is important for $NO_2$ analysis, please clarify this point.

**Response**: For this paper all TROPOMI SNO2 observations that are sufficiently accurate (qa value > 0.5) are included in the averaging at the 0.4×0.8 degree grid. Note that the qa_value threshold of 0.5 is defined such that effectively only measurements with SZA > 81.2 degrees are excluded. Overall it means that data from ascending and descending orbits are used in the calculation of daily mean TROPOMI SNO2. The average is arithmetic without any weighting.

We also added a brief discussion on SNO2 diurnal cycle effects on both TROPOMI SNO2 retrievals and validation results. Although this is not expected to materially affect the results of this paper there are clear indications that effects are not marginal. Which supports the need for an assessment of diurnal cycle effects. As we present a more-or-less new SNO2 application for a region (Polar) that otherwise has been largely ignored there is little information – if any – about diurnal cycle effects. What has been published indicates what effects can be sufficiently large that they cannot be simply ignored but not large enough to materially affect the results presented here.

Note in support that the 10-20% SNO2 diurnal cycle adjustment effects reported in Dubé et al. [2021] are consistent with the SAOZ 10% SNO2 diurnal cycle correction effects mentioned in Compernolle et al. [2021].

**Action**: the following was added to the discussion section 4.

"In addition, although the diurnal cycle in SNO2 is relatively small compared to its seasonal cycle it nevertheless can affect satellite retrievals and validation results. Dube et al. [2021] reported order of magnitude 10-20% effects for SAGE III/ISS solar occultation limb retrievals with larger effects for higher latitudes. Although their results are not one-on-one applicable to the results presented here they clearly indicate the need for properly assessing diurnal cycle effects on TROPOMI SNO2 measurements and validation."

We also added the following to the end of section 2.1.

"A qu_value > 0.5 excludes any TROPOMI observation with a solar zenith angle > 81.2°. During the Antarctic summer this leads to some observations from the descending TROPOMI orbit over Antarctic to be include in the daily average (TROPOMI orbits the sunlit part of the earth from south to north)."

3) The authors use the term "Noxon cliff" for both the cliff for NO2 and that of O3. However, as far as I understand, the "Noxon cliff" can be used only for the cliff of nitrogen species (NOx, HNO3, N2O5, etc.), but not for O3. Therefore, all the description after Section 3.4, where the authors use the terminology "Noxon cliff for TCO3" should be re-worded.

**Response:** we agree, originally the Noxon cliff was indeed associated with nitrogen oxides and the strong-cross-vortex-edge gradient that was observed in nitrogen oxides. We do want to note in passing that several subsequent publications have connected the Noxon cliff to similar cross vortex trace gas gradients observed for other trace gases. Which should not surprise anyone as there are other trace gases involved in the catalytic ozone destruction cycle that will show strong similar gradients (HOx, CLOx or BROx cycle gases) while the mixing barrier across the winter vortex may also lead to significant gradients in other trace gases not involved in catalytic ozone destruction cycles.

**Action:** changed the description to "cross-vortex TCO3 gradient" (or similar) and checked/ensured that the use of the phrase "Noxon cliff" was exclusively used in conjunction with stratospheric nitrogen cycle trace gases.

**Minor Comments**

4) P.6, Figure 1A: Please explain why there are differences in darkness both in S5p total NO2 data points and reference total NO2 data points in this plot.

**Response**: the data points are semi-transparent circles (with darker outline) to provide the reader with some idea of where TROPOMI and SOAZ data overlap. Due to the strong seasonal cycle and relatively small differences between both datasets points frequently overlap. The plots on the S5p VDAF server consist of non-transparent filled markers. This has the consequence that many data points become invisible - either overlapping data points from the same instrument or overlapping TROPOMI and SAOZ data points. We thought that using semi-transparent data points was visually a bit more appealing. The consequence is that overlapping data points will show up with a different transparency. Note that for each SAOZ data point there is a corresponding TROPOMI data point.

**Action:** We added a clarification to the figure caption.

5) P.6, Figure 1B: There is no explanation on different three regression lines. Please explain them either in caption or in the legend. Also, no color bar is shown in the figure. Please add a color bar.

**Response:** we indeed forgot to describe which line is which (although the color of the value of the regression coefficients provide in the upper left corner of the plot provide a visual clue). The dotted grey line is the 1:1 line, the solid grey line is the Ordinary Linear Regression line, the solid red line is the Orthogonal Distance Regression line.

**Action:** We added a clarification to the figure caption.

6) P.6, Table 1: Why there are relatively large biases (< -10 %) in Rio Gallegos data? Please add some explanation.

**Response:** Comparison between TROPOMI and Pandora total NO2 column data and separating stations by (tropospheric NO2) pollution levels reveals a systematic small positive "bias of +5.8 % for the 28 lower polluted stations and -17.9 % for the 42 higher polluted stations" (ground-based measurements larger than TROPOMI), see Lambert et al. [2023]. This negative bias for polluted stations is consistent with the negative bias for Rio Gallegos and of similar order of magnitude.

Although the Pandora network does not cover high southern latitudes, the possibility exists that Rio Gallegos measurements are affected by air pollution from the city of Rio Gallegos itself (population of approximately 80,000). The location of the SAOZ instrument at "Observatorio Atmosférico de la Patagonia Austral" is west-north-west to the city and bordering the airport (see Google Maps image below). The physical distance to the city border is approximately 5 km and to the city center approximately 10 km, sufficiently nearby for combustion NO2 from the city to affect the SAOZ observations.

The validation report by Lambert et al. [2023] does not find biases due to the satellite solar zenith angle (SZA), the satellite cloud fraction and satellite surface albedo large enough to explain the relatively large bias for Rio Gallegos. Note that Lambert et al. [2023] does not specifically discuss comparisons for individual locations.

An analysis of SAOZ NO2 data at Rio Gallegos by Raponi [2012] reveals that the lower envelope of the NO2 seasonal cycle is well and sharply defined – suggesting a clean troposphere with the stratospheric seasonal cycle dominating. The upper envelope, however, shows a lot of scatter and spikes – which are absent at clean Southern Hemisphere locations like Neumayer suggesting emission plumes passing over the SAOZ station under favorable conditions. Note that Raponi [2012] does not discuss the causes of these spikes.

Overall, contamination of Rio Gallegos measurements by local tropospheric pollution would be a possible and plausible explanation but would require more research. Note that even with the bias the difference remains within the TROPOMI mission requirement targets.

**Action:** a brief summary/explanation based on the response above was added to section 2.2

*reference*

*Measurements of NO2 and O3 vertical column densities over Río Gallegos, Santa Cruz province, Argentina, using a portable and automatic zenith-sky DOAS system*

*Optica Pura y Aplicada 45(4):397-403*

*DOI:10.7149/OPA.45.4.397*

*https://www.researchgate.net/publication/272963549_Measurements_of_NO2_and_O3_vertical_column_densities_over_Rio_Gallegos_Santa_Cruz_province_Argentina_using_a_portable_and_automatic_zenith-sky_DOAS_system*

[Figure]

7) P.6, Table 1: The order of stations in the table should be not in alphabetical order, but from lower latitude to higher latitude.

**Action:** changed accordingly

8) P.7, L.195: "MSR-2" first appeared in the text which is not explained elsewhere, nor any reference is shown. Please explain MSR-2 and add some references.

**Action:** a brief description of MSR-2 and some references were added to section 2.3 (Global Ozone field data).

9) P.7, L.197: "TEMIS" first appeared in the text which is not explained elsewhere, nor any reference is shown. Please explain TEMIS and add some references.

**Action:** we removed "TEMIS" and rather refer to the "KNMI operational daily global assimilated TCO3 field". We also added that this operational assimilated TCO3 field is produced for operational daily worldwide UV index predictions and that these TCO3 analyses and predictions – input for the UV index predictions - are always in real time available – unlike MSR-2 which is updated once a year or so.

10) P.7, L.203: The authors claimed "longitude-latitude grid of 1.5°x1.0° and is re-gridded to 0.8°x0.4° to match …". How they re-grid the data into finer resolution grid? Please explain.

**Response:** This is correct, the TCO3 data is regridded to a finer resolution using a standard bilinear interpolation. Obviously it could have been decided to retain the lower TCO3 resolution and average the TROPOMI NO2 data to on that grid. TROPOMI NO2 data has a much finer resolution (3.5x5.5 km sub-satellite) so the 0.8x0.4 grid already involves and averaging step. The 0.8x0.4 grid is then somewhere between the TROPOMI NO2 resolution and the TCO3 resolution. This could have been done differently but each choice comes with its pros and cons. We did, however, check for a single day what results looked like using TROPOMI TCO3 pixel data – so at a spatial resolution similar to TROPOMI NO2 – and we did not find significant differences in the overall outcomes (this is mentioned at the end of discussion section 4): "results … are robust relative to using gridded data or pixel data or even data from different satellites". Obviously this is a point for further attention in the future but sufficient for the purpose of this paper.

**Action:** we changed this to "re-gridded (bi-linear interpolation) to a finer 0.8 ..."

11) P.7, L.205: "GOME-2 has a 4 DU bias". Is this a positive bias or a negative bias? Please explain.

**Response:** The bias (offset) is positive relative to ground-based observations (see van der A et al., 2015)

**Action:** the text was modified accordingly

12) P.8, Figure 4: The panel for 2020 is wrong (the one shown here is for 2019). Please add a panel for 2020.

**Action:** figure was updated

13) P.8, Figure 5: Please add panel numbers [A]-[F] in Figure 5. In the figure caption, use [A]-[F] for the corresponding explanation.

**Action:** figure was updated and the text modified accordingly

**Grammar/Typos**

**Action:** all grammar issues and typos have been changed accordingly

14) P.1, L.28: This process depletes the Antarctic … --> This process depletes nitrogen oxides (denitrification/denoxification) in the Antarctic stratospheric vortex (Farman …

15) P.2, L.29: Farman et al., 1995 --> Farman et al., 1985

16) P.2, L.31: during Antarctic spring to the then denitrified … --> during Antarctic spring to the denitrified …

17) P.2, L.45: Struthers et al 2004 --> Struthers et al., 2004

18) P.6, L.170: regression coefficients equal 0.94 --> regression coefficients equal to 0.94

19) P.11, L.336: And complex relationships between (long-lived) … --> And complex relationships among (long-lived) …

20) P.14, L.416: J d.L. --> A.d.L.

21) P.15, L.417: P.V. --> J.P.V.

---

## Author Comment (AC2)

**RC2**

**General comments:**

"The Antarctic stratospheric Nitrogen Hole: Southern Hemisphere and Antarctic springtime total nitrogen dioxide and total ozone variability in Sentinel-5p TROPOMI data" provides a scientifically useful analysis of a TROPOMI measurements of nitrogen dioxide during the Antarctic ozone hole. The study demonstrates that co-located TROPOMI $NO_2$ and $O_3$ observations can be used to clearly identify the evolution of chemical differences between inner and outer polar vortex air masses throughout the springtime. While demonstrating the viability of a new dataset for the analysis of the Antarctic ozone hole is scientifically important, improvements to the presentation of the data and details about the design of the analysis are needed.

**Response:** we thank the referee for the review efforts and the comments that have helped improve the paper. Below follows a detailed response to the comments including for each comment a description of the modifications that have been made.

**Specific comments:**

1. The introduction could be improved by focusing on the advances offered by the TROPOMI dataset and the authors' analysis, specifically:
   a. What improvements or unique capabilities does this satellite dataset offer?
   b. What problem or scientific question does the dataset answer?
   c. The background on prior satellite studies could be condensed.

**Response**

**1a.** The TROPOMI satellite instrument builds on the legacy of hyperspectral UV/VIS satellite instruments like GOME, SCIAMACHY, OMI, and GOME-2. TROPOMI provides satellite observations with unprecedented spatial resolution, accuracy and precision compared to its predecessors. Although designed for improved monitoring tropospheric pollution, it nevertheless also equally improves stratospheric NO2 and/or total NO2 column observations.

**Action:** we add the following to the introduction:

"The TROPOspheric Monitoring Instrument (TROPOMI) is the first of the next generation of hyperspectral UV/VIS satellite instruments. Designed and developed based on experience with satellite instruments like GOME, SCIAMACHY, GOME and GOME-2 it provides satellite observations of unprecedented spatial resolution and accuracy."

**1b.** Apart from providing measurements of a trace gas relevant for stratospheric ozone and catalytic ozone depletion at an unprecedented scale and with unprecedented accuracy (as explained in **1a**), a key aspect is that these measurements add some new stratospheric monitoring capacity that otherwise currently is suffering from aging satellites and steady decline in the number of such satellites. The scientific community has a commitment towards continued monitoring of the stratosphere and the ozone layer as part of the Montreal Protocol "for the protection of the ozone layer". Satellite measurements have been crucial for this commitment. Fewer satellite and deteriorating satellites are making meeting this commitment more difficult. Developing new satellite data applications is therefore very much welcomed. Especially is they are based on satellite instruments that are expected and planned to be operational for many decades ahead in time.

**Action.** We added the following paragraph to the introduction:

"Furthermore, the current suite of satellites that can be used for stratospheric monitoring is aging and the number of such satellites is dwindling. This is a significant concern for the scientific community and their commitment towards monitoring the ozone layer as part of the Montreal Protocol for "Protection of the Ozone Layer". Recovery due to the phase out of emissions of ozone depleting substances is a slow process and full recovery is only expected in the second half of the 21$^{st}$ century. However, unusual stratospheric events can strongly affect the ozone layer thickness from year to year. Whether such year-to-year changes in stratospheric ozone are anomalous or the result of natural variability is crucial for confident statements whether recovery is progressing as expected (or not). Satellite instruments measuring the stratospheric chemical composition other than ozone have been essential for understanding this year-to-year variability and thus meeting the commitment of the scientific community towards monitoring the ozone layer support of the Montreal Protocol. Given the aging suite of stratospheric monitoring satellites and their dwindling numbers, identifying new stratospheric monitoring applications is more than welcome for continued stratospheric monitoring. Especially if these applications are based on satellite instruments that are planned to remain available for many decades into the future."

**1c.** Because the topic and application introduced in this paper in essence is new – with the exception of the two exploratory research papers based on the earliest generation of hyperspectral UV/VIS satellite instruments in the mid-2000s – our thinking was to provide an as complete as possible earth observation context of stratospheric NO2 observations - taking limitations in publication length into account.

Hence the lengthy "background on prior satellite studies". We are pretty confident that knowledgeable people in the field will ask – just like we did – as to why this has not been published before and how this related to the extensive research done on stratospheric chemistry.

Furthermore, given that this paper could be the start of a series of new publications an overview with a "background on prior satellite studies" would provide a good starting point for anyone building on our paper.

Hence out preference to keep the section on the "background on prior satellite studies". We could do otherwise but would prefer for the editor to make a decision on this.

**Action:** we leave it up to the editor to decide if the "background on prior satellite studies" should be shortened or not.

2. Additional details in the methods describing the range of latitudes used in each analysis are needed.

**Response**. See comment (2) by referee #1 for a similar question and our response that comment.

**Action:** the following was added to the discussion section 4.

"In addition, although the diurnal cycle in $SNO_2$ is relatively small compared to its seasonal cycle it nevertheless can affect satellite retrievals and validation results. Dube et al. [2021] reported order of magnitude 10-20% effects for SAGE III/ISS solar occultation limb retrievals with larger effects for higher latitudes. Although their results are not one-on-one applicable to the results presented here they clearly indicate the need for properly assessing diurnal cycle effects on TROPOMI $SNO_2$ measurements and validation."

We also added the following to the end of section 2.1.

"A qu_value > 0.5 excludes any TROPOMI observation with a solar zenith angle > 81.2°. During the Antarctic summer this leads to some observations from the descending TROPOMI orbit over Antarctic to be include in the daily average (TROPOMI orbits the sunlit part of the earth from south to north)."

3. In the analysis of $NO_2/O_3$ correlations, how are the mask boundaries determined? What effects are there, if any, on the conclusions of the analysis if the boundaries are varied?

**Response:** the boundaries between the three different masks are the result of a manual iterative process. We came to these boundary after some experimenting and testing. The main reason for them is that these three masks consistently separate the inner-edge-outer vortex regions, regardless of the time period start, the time period end and the timp period duration. In that sense these boundaries are robust, although they are purely empirical. We have entertained the thought to distinguish between two distributions in vortex edge region (see answer to next question) but decided against it to keep it simple.

**Action:**

4. Can the "mixing lines" be geographically and temporally isolated into discrete eddies/filaments?

**Response:** this continues/builds on the answer to comment (4) above.

In short: good question, in all honesty we don't really know, it surely looks like it could be the case but that requires much more research.

There is at least an easy separation possible for the MASK-3 region – as is also obvious from the phase diagrams. There is almost always a vortex edge section with reduced NO2 and reduced O3. This is likely explained by total ozone over Antarctic during Antarctic springtime generally following a wave-1 pattern (maximum-minimum-maximum along a full longitude circle). The location of the minimum and maximum rotates throughout spring in a clockwise fashion and with a time scale of two weeks or so (if I recall correctly). We believe that these locations might be areas of something one could call "vortex leakage", areas where cross-vortex edge mixing takes place

Given the multiple TROPOMI overpasses per day – including ascending and descending orbits – could be the reason that during daytime there appear multiple "mixing lines" in the phase diagram (Figure 5). Simply a case of the vortex edge regions having spatially moved during the 12 hours between the previous and next overpass during late Antarctic spring. For the phase diagrams covering longer periods it becomes more and more difficult to distinguish these mixing lines, as evidence by many figures in the paper. Rather they developing into a continuous distribution: the vortex edge continuously changing locations and thereby NO2-O3 ratios due to different dynamics and advection and different altitudes.

However, to fully understand and explore vortex edge dynamics requires much more extensive research, something we felt more suitable for a follow-up paper and/or research proposal. That includes the question whether mixing lines could be "geographically and temporally isolated into discrete eddies/filaments". Like stated above, yes, a possibility, looks like it, but for now we don't know. Furthermore, we have done much more analysis and made many more plots than ever would be possible to show in a paper (see as an example the animation in the supplement) so we already had to condense a lot of material for this paper.

**Action:** no action

        a. Can the simple phase analysis of "mixing lines" recreate a similar structure on a 2-d plot? Figure 9 is not convincing on its own. Consider including actual seasonal trends.

**Response:** see also previous answer. We are not sure what the referee means here: the aim of this conceptual figure is to highlight that shifts in (spatial) phasing in even simple distributions with different locations spatially of minimum and maximum values lead to variations in a phase-diagram that is not dissimilar from what is observed.

We constructed Figure 8 based on an extensive series of TROPOMI SNO2-TCO3 phase diagrams along a range of latitudes and for a range of dates that show more or less similar albeit more complex patterns that nevertheless in essence come down to the same point: phase differences in collocated data with different locations of their minimum and maximum.

But as in previous answers: space limitations in a research paper do not allow us to present most of the analyses and figures we made.

Hence why we prefer to keep Figure 9 as it is.

Note that obviously latitude is not the best coordinate here – equivalent latitude would be preferred but that is also something for a future project.

**Action:** no action

**Technical comments:**

1. Subscripts for the common notation of $SNO_2$ and $TCO_3$

**Action:** changed through the document

2. Improve the figure labels, especially dates

**Action:** figures have been updated also in accordance with a number of comments from referee #1

3. Figure 4 is redundant

**Response:** without further explanation it is not clear why figure 4 is redundant. We would argue Figure 4 provides necessary spatiotemporal global (zonal mean) information on the seasonal cycle in SNO2. No other figures contain similar information. Also, as this paper are in part is inspired by Wespes et al. (2022) who present a similar figure for IASI HNO3 this allowing a visual direct comparison with their results (the TCO3-SNO2 phase-diagrams are something new and not available for other trace gas combinations).

Hence we have a strong preference to keep Figure 4.

---

## Referee Report (RR1)

2nd Review of "The Antarctic stratospheric Nitrogen Hole: Southern Hemisphere and Antarctic springtime total nitrogen dioxide and total ozone variability as observed in Sentinel-5p TROPOMI data" by A. de Laat et al.

< General Comments >

I felt that the authors have made great effort to revise the paper. I think that most points which I have raised have satisfactorily answered by the authors. I think that the paper is now ready to be published. I have some minor comments which had better modify before publication which is described below.

< Major Comments >

1) In the RC1 1), change of the action is suggested. However, in the revised manuscript, the following sentence is somehow missing. I would recommend to add it to the manuscript. "The amplitude of the adjustment depends strongly on the effective SZA assigned to the ZSL-DOAS measurement: it is taken here to be 89.5.

2) Section 2.3, the authors added a references [van der a et at, 2010, 2015], which are not found in the reference list.

3) Figure 1A. The last sentence ends somewhat strange: "Note that for …". Please add correct catption.

---

## Editor Decision (ED1)

**Comments on egusphere-2023-2384-revised**

de Laat et al., The Antarctic stratospheric Nitrogen Hole: Southern Hemisphere and Antarctic springtime total nitrogen dioxide and total ozone variability as observed in Sentinel-5p TROPOMI data

Title: change "in Sentinel-5p TROPOMI data" to "by Sentinel-5p TROPOMI".

P1, L8: "Denitrification of the stratospheric vortex" -> I would rather write "within in the stratospheric vortex" or "of the stratosphere".

P1, L10: You should add a sentence and explain the connection between denitrification and the Nitrogen Hole.

P1, L14: "are what is" -> rather "is what is"? Anyway I would suggest to rephrase the sentence since both "is what is" or "are what is" does not sound so nice.

P1, L21: "is extended with past satellite observations" -> why only past? What about future satellite observations? These would also be valuable.

P1, L26-27: Rephrase sentence? Sounds that there is something missing. Maybe "that occur" after Antarctic winter.

P1, L27: Particles themself are no clouds. These form clouds. Thus, I would suggest to write "forming so-called polar stratospheric clouds – whose particles sediment after they have grown large enough for gravitational settling".

P1, L28: Move "(denitrification/denoxification)" behind process and write "called denitrification/denoxification".

P1, L23ff: You focus in your description solely on the Antarctic (which is reasonable since your study focuses on the Antarctic), but the processes described hold for both hemispheres and also the given references are for both hemispheres. Thus, I would suggest to rewrite the text and describe these processes naming both hemispheres (or call it just polar regions). You only need to be careful with the term "ozone hole" since this only holds for the Antarctic.

P2, L30: Please rephrase this sentence. It is not only the "presence", rather the strong border of the vortex.

P2, L35: rather "exist" than "re-formate".

P2, L37: Please rephrase this sentence. The vortex breakup has nothing to do with the increasing sunlight and absorption. The vortex break up or instability is always coupled to dynamical processes (waves, wind reversal from winter to summer conditions).

P2, L50: Please state instead of "mostly Antarctic" clearly "Arctic and Antarctic".

P2, L66: Please rephrase this sentence. You shouldn't write it like this, thus accusing the authors of mentioning something, but not showing it". It is quite reasonable that studies focus on one hemisphere although the processes investigated are found in both hemispheres.

P3, L68, 72 and 80: Instrument names like "GOME", "OMI" and "IASI" should be introduced.

P3, L70: What is SNO2? Stratospheric NO2? The abbreviation should be introduced.

P3, L71: is -> has been

P3, L77: Remove "but do not analyze those observation in more detail." Don't point out what others have not done, solely focus on what has been done previously.

P3, L79: add "in" -> in satellite nadir observations

P3, L81:The removal of HNO3 is the main process (and not part) of the denitrification since PSC particles contain HNO3. Please rephrase the sentence.

P3, L92: Introduce abbreviation "SCIAMACHY".

P4, L103: I would suggest to rephrase to e.g.: "This makes the TNO2 or SNO2 from TROPOMI particularly suitable for explaining the Noxon cliff………Further, you should emphasize here that TROPOMI has higher resolution compared to other nadir instruments.

P4, L117: In this paragraph references should be added. For the statements about ozone you could use the WMO report.

P5, L132: I am myself no sure, but I would rather replace "in" by "from" or write "performs measurements at four channels".

P5, L138: wide -> width

P5, L144: What is TMP-MP. Abbreviation should be introduced.

P5, L149: Add "Antarctic" before "NO2".

P6, L161: qu_value -> qa_value?

P6, L163: include -> included

P6, L166: Section 4 -> Sect. 4

Note, generally in Copernicus journals Figure and Section are abbreviated as Fig. and Sec. respectively, except when they appear at the begin of the sentence. Check ACP manuscript preparation guidelines and adjust this in the manuscript accordingly.

P6, L179: Parenthesis instead of brackets.

P7, L193: means -> mean

P7, L198: appendix -> Appendix

P7, L212: Parenthesis instead of brackets.

P8, L241: Please introduce the abbreviation "TEMIS".

P8, L250: spatiotemporal -> Spatiotemporal

P9, L267: ….outside the Antarctic vortex ……kept out of the vortex …….-> isn't that some kind of doubling? Please rephrase.

P9, L270: for -> from

P9, L274: I guess MASK-1 refers to inner vortex, MASK-2 to outer vortex and MASK-3 to vortex-edge. Add this to the sentence (put the respective MASK behind the respective vortex area).

P10, L285: multi-day -> Multi-day

P10, L287: replace "-" by to, so that it reads "(5-10 to 15-30 days)".

P12, L350: there still is well defined ->  there is still a well defined

P12, L355: 2018-2020-2021 -> 2018, 2020 and 2021

P13, L386: Introduce instrument names "OSIRIS", "ACE-FTS" and "MAESTRO".

P13, L395: What is the abbreviation QA4ACV standing for? Please add.

P13, L397: Introduce "OMPS".

P13, L410: Parenthesis instead of brackets.

P14, L417: appendix -> Appendix

P14, L435: Ozone Hole -> ozone hole

P14, L444: 2018-2020-2021 -> 2018, 2020 and 2021

P15, L460-471: Check line spacing. Should be the same as for the entire manuscript.

P17 and 18: Combine these two figures either in one Figure (with one caption) or number them as Figure 1 and 2.

P18, L493: appendix -> Appendix

P21, Figure 4 and all other figures: Use parenthesis instead of brackets for the panels. This holds also for the axis labeling.

P21, Figure 4: I would suggest to also mark the stations in the other panels.

P25, Figure 5 and all other figures: parenthesis instead of brackets and small instead of capital letters should be used.

P29, L529: add degree sign so that it reads 30°S and 90°S.

P30, L541: Add "correlation" before "coefficient".

P31ff, References: Check all references add adjust to Copernicus style. Journal names should be abbreviated accordingly and titles should be in a common manner and the subscripts for the numbers in the names of the chemical species should be used.

P44 and P45, figure captions: table -> Table

---

## Author Response (AR2)

We have addressed the questions and suggestions as outlined below.

We have indicated a handful of points (underlined in bold) that the editor may want to double check with one point we would like to keep as is and a modification we made that was not requested but we nevertheless thought useful:

- With regard to discussing the two 20-year old exploratory papers on this topic and discussing what was and was not done in those papers [point 13 below] we kindly ask to keep it as is. The statement about what was not done was not meant as criticism but merely factual and something we believe certain readers may want to know.

- We modified Appendix Fig. A3 which originally had the same layout as Fig. 5 but we realized that except for Fig. 5a all differences were marginal (which was the point of A3 anyway), hence just showing two versions of that panel (a) of Figure 5 suffices.

We hope that we have therewith addressed all issues raised.

**Response to editor comments**

[0] Title was changed as suggested

[1] We merged figures 1A and 1B with two panels (A+B) and one figure caption.

[2] P1, L8: added "Sedimentation of large nitric acid trihydrate polar stratospheric cloud particles within the Antarctic polar stratospheric vortex that form during winter deplete the inner vortex from nitrogen oxides"

[3] P1, L14: modified the sentence to "Connecting both main regimes is a third regime of coherent patterns in the total nitrogen dioxide column - total ozone column phase space – defined here as "mixing lines"."

[4] P1, L12: changed to "and long term monitoring of Antarctic Ozone Hole conditions."

[5] P1, L26-27: changed "after formation of" to "and the development of" (Extremely low stratospheric temperatures during Antarctic winter and the development of the stratospheric polar vortex results in …)

[6] P1, L27: modified as suggested

[7] P1, L28: modified as suggested

[8] P1, L23ff: we changed the description from "Antarctic" to "polar" although we still have to make the switch to the Antarctic as that is where we analyze data in this paper. **We hope this meets the request but please do check.**

[9] P2, L30: changed to "Strong zonal winds at the stratospheric polar vortex edge prevent …"

[10] P2, L35: changed to "… and unfavorable for PSCs while favorable again for stable halogen reservoir species like HCl …" **→ please check, we found it not easy to come up with a satisfactory sentence.**

[11] P2, L37: This was not meant as a reference to the vortex breakup but rather the association with increase planetary wave activity. We changed it now to "but sometimes also by increased

planetary wave activity (de Laat and van Weele, 2011; Wargan et al., 2020; Smale et al., 2021)".
**Note that we wanted to keep the explanation short, hence why we only refer to papers already in the reference list (because there are plenty more possible references).**

[12] P2, L50: we alternatively deleted "mostly Antarctic". The sentence originally read "(mostly Antarctic) polar …". Polar already indicates that it can be either Arctic or Antarctic, hence there is no reason to add "(Arctic and Antarctic) polar". Even though most studies into the Noxon cliff have focused on the Antarctic.

[13] P3, L66: **if the editor allows us we would rather keep this in this particular case because we think it is important: looking in hindsight we found it very surprising that no one had looked into this for the past 20 years. Especially after the two initial explorative papers we discuss here. Although we have our ideas about why this has been left. The description here is not (meant as) an accusation but a factual brief summary of what was shown and discussed (and not). We expect that is of interest to some readers (we thought it interesting to note this as it clearly puzzled us).**

[14] P3, L68: full instrument names of GOME, OMI and IASI are added

[15] P3, L70: SNO2 abbreviation was already introduced in L63 (and yes = stratospheric NO2)

[16] P3, L71: changed

[17] P3, L77: deleted

[18] P3, L81: changed to "is the main denitrification process"

[19] P3, L91: m

[20] P4, L103: changed to "This makes TROPOMI $TNO_2$ or $SNO_2$ – in particular combined with the much higher spatial resolution of TROPOMI compared to OMI, GOME-2 and the Ozone Mapping and Profiler Suite (OMPS) - particularly suitable for exploring the Noxon Cliff."

[21] P4, L117: added a reference to "(WMO 2022)"

[22] P5, L22: we do indeed think "from" is better here

[23] P5, L138: change to "width"

[24] P5, L144: changed to "(for TROPOMI: TM5-MP - Transport Model version 5 – Massive Parallel; Williams et al., 2017)", note the added reference

[25] P5, L149: changed to "nearly all Antarctic"

[26] P6, L161: correct, should read "qa_value"

[27] P6, L163: changed

[28] P6, L166: changed

[29] : changed Figure and section to Fig. and Sec. in the entire document

[30] P6, L179: changed

[31] P7, L193: changed

[31] P7, L198: changed

[32] P7, L212: changed

[33] P8, L241: "Tropospheric Emission Monitoring Internet Service"

[34] P8, L250: changed

[35] P9, L267: chanted to "springtime advection of SNO2 enhanced stratospheric air into the Antarctic stratospheric vortex is limited during …"

[36] P9, L270: "found for" change to "from"

[37] P9, L274: changed

[38] P10, L285: changed

[39] P10, L289: changed

[40] P12, L350: changed

[41] P12, L355: changed

[42] P13, L386: changed

[43] P13, L395: changed: QA4ECV = Quality Assurance for Essential Climate Variables

[44] P13, L397: changed

[45] P13, L410: changed

[45] P13, L417: changed

[46] P14, L435: changed

[47] P14, L444: changed

[48] P15, L460-471: checked

[49] P17-P18: figures A1 and 1B are now combined in one figure with two panels (1A + 1B)

[50] P18, L493: changed

[51] Figure 4: changed to using parenthesis instead of brackets for all relevant figures and switched from upper case to lower case letters. Checked the document for referencing to panels (lower case instead of upper case). **Modified Appendix Fig. A3 which was similar to Fig 5. but now only showing and comparing Fig. 5a rather than all panels of Fig. 5. Appendix Fig. A3 showed the results of Fig 5. But for TROPOMI SNO2 data at pixel level rather than averaged at 40x80 km. As differences are marginal, we thought there was no added value of showing all Fig 5. panels in Appendix Fig. A3. The comparison with Fig. 5a – 2D histogram – should be sufficient to convince the reader that for the results presented in this paper there is no real material difference in using TROPOMI SNO2 at 40x80 or at the TROPOMI pixel level.**

[52] Figure 4: added stations to Figs. 3 and 5 (note that it is not needed for Fig. A3 due to the modification as described in the answer to [51])

[53] Figure 5: see answer [51]

[54] P29, L529: changed

[55] P30, L541: changed

[56] P31ff: checked to the best of my abilities

[57] P44, P55: changed

**Response to Referee #1**

1) In the RC1 1), change of the action is suggested. However, in the revised manuscript, the following sentence is somehow missing. I would recommend to add it to the manuscript. "The amplitude of the adjustment depends strongly on the effective SZA assigned to the ZSL-DOAS measurement: it is taken here to be 89.5.

Correct and well spotted, the sentence "it is taken here to be 89.5" has been added.

2) Section 2.3, the authors added a references [van der a et at, 2010, 2015], which are not found in the reference list.

In both submitted revision versions (with/without track changes, on the COPERNICUS webportal, submitted files from 26 January) both "van der A et al. [2010, 2015]" papers are in the reference list. The 2015 paper has always been in the reference list, the 2010 paper was added after the revision.

Probably what has happened is that the referee looked under the "V" whereas we listed both papers under "A" (always a problem with Dutch last names).

3) Figure 1A. The last sentence ends somewhat strange: "Note that for …". Please add correct catption.

Turns out that the rest of the sentence is on the next page. We fixed that in the draft paper but for the eventual edited document that does not matter.